# The Effects of Major Mushroom Bioactive Compounds on Mechanisms That Control Blood Glucose Level

**DOI:** 10.3390/jof7010058

**Published:** 2021-01-16

**Authors:** Jelena Aramabašić Jovanović, Mirjana Mihailović, Aleksandra Uskoković, Nevena Grdović, Svetlana Dinić, Melita Vidaković

**Affiliations:** Department of Molecular Biology, Institute for Biological Research “Siniša Stanković”, National Institute of Republic of Serbia, University of Belgrade, Bulevar Despota Stefana 142, 11060 Belgrade, Serbia; mista@ibiss.bg.ac.rs (M.M.); auskokovic@ibiss.bg.ac.rs (A.U.); nevenag@ibiss.bg.ac.rs (N.G.); sdinic@ibiss.bg.ac.rs (S.D.); melita@ibiss.bg.ac.rs (M.V.)

**Keywords:** diabetes mellitus, mushroom, polysaccharides, terpenoids, anti-hyperglycemic effects

## Abstract

Diabetes mellitus is a life-threatening multifactorial metabolic disorder characterized by high level of glucose in the blood. Diabetes and its chronic complications have a significant impact on human life, health systems, and countries’ economies. Currently, there are many commercial hypoglycemic drugs that are effective in controlling hyperglycemia but with several serious side-effects and without a sufficient capacity to significantly alter the course of diabetic complications. Over many centuries mushrooms and their bioactive compounds have been used in the treatment of diabetes mellitus, especially polysaccharides and terpenoids derived from various mushroom species. This review summarizes the effects of these main mushroom secondary metabolites on diabetes and underlying molecular mechanisms responsible for lowering blood glucose. In vivo and in vitro data revealed that treatment with mushroom polysaccharides displayed an anti-hyperglycemic effect by inhibiting glucose absorption efficacy, enhancing pancreatic β-cell mass, and increasing insulin-signaling pathways. Mushroom terpenoids act as inhibitors of α-glucosidase and as insulin sensitizers through activation of PPARγ in order to reduce hyperglycemia in animal models of diabetes. In conclusion, mushroom polysaccharides and terpenoids can effectively ameliorate hyperglycemia by various mechanisms and can be used as supportive candidates for prevention and control of diabetes in the future.

## 1. Introduction

Diabetes mellitus, characterized by hyperglycemia (abnormally elevated fasting and postprandial glucose level in the blood), represents a heterogeneous group of disorders of multiple etiologies and a major health problem worldwide. While great effort has been made in the management of diabetes, its prevalence continues to grow together with a serious increase of morbidity and mortality related to the development of diabetic complications [1]. While chemical and biochemical hypoglycemic agents, e.g., insulin, tolbutamide, phenformin, troglitazone, rosigitazone, and repaglinide, are the mainstay of treatment of diabetes and are effective in controlling hyperglycemia, they have harmful side-effects and fail to significantly alter the course of diabetic complications [2]. Mushrooms, traditionally used as remedies for diabetes healing, represent an exciting field for developing new types of therapeutics to control diabetes and its complications. Some mushrooms have demonstrated clinical and/or experimental control of blood glucose and modification of the course of diabetic complications without side-effects [3,4]. To date, more and more bioactive components including polysaccharides and their protein complexes, dietary fibers, terpenoids, and other compounds extracted from fruiting bodies, cultured mycelium, or cultured broth of medicinal mushrooms have been reported to have anti-hyperglycemic activity. These compounds exhibit their antidiabetic activity via different mechanisms. However, there is insufficient scientific or clinical evidence to draw definitive conclusions about the efficacy and safety of individual medicinal mushrooms or their isolated bioactive compounds to be used as official drugs for treatment of diabetes. Therefore, well-designed randomized controlled trials with long-term consumption are needed to guarantee the bioactivity and safety of mushroom products for diabetic patients. This review covers contemporary drug therapy for diabetic patients and underlines recent studies that demonstrated the hypoglycemic effect of mushroom major bioactive components as well as the importance of their use in the treatment of diabetes. In addition, known mushroom bioactive component-induced mechanisms and pathways involved in lowering blood glucose concentration are elaborated.

## 2. Diabetes—Sweet and Silent Killer but Not Unbeatable

Diabetes is life threatening and one of the most common chronic diseases worldwide that has reached alarming levels. Today, according to the International Diabetes Federation nearly half a billion people are living with diabetes worldwide. This number is predicted to increase dramatically, reaching 578 million by 2030, and 700 million by 2045 [5], confirming that diabetes is one of the fastest growing global health problems nowadays.

Diabetes mellitus is a long-term metabolic disorder characterized by elevated blood glucose concentration that results from absolute insulin deficiency or insufficient insulin secretion and/or insulin sensitivity [6]. According to Schmeltz and Metzger [7], a new classification of diabetes relies on etiology and pathophysiology, without distinction regarding age of onset or type of treatment. The main categories of diabetes are type 1, type 2, double (hybrid), and gestational diabetes mellitus.

Type 1 diabetes results from autoimmune destruction of the pancreatic β-cells [8,9], which usually leads to absolute insulin deficiency. Patients with this type of diabetes require insulin therapy to maintain normoglycemia and a healthy lifestyle to manage their condition effectively [10]. Type 1 diabetes which usually affects children or young adults accounts for 5–10% of the total diabetic population.

Type 2 diabetes may range from predominant insulin resistance with relative insulin deficiency to a predominantly insulin secretory defect with insulin resistance [10]. This type of diabetes is most commonly seen in older adults and accounts for 90–95% of the total diabetic population. The major risk factors for type 2 diabetes are family history of diabetes, age, obesity, unhealthy diet, ethnicity, and physical inactivity [11]. Nowadays, type 2 diabetes can be increasingly seen in children, adolescents, and younger adults due to an unhealthy way of life. The symptoms are often unobvious because the blood sugar is not high enough to be noticed. In most cases, the disease is diagnosed many years later when hyperglycemia together with macrovascular and microvascular complications becomes apparent.

Since precise definition and diagnosis of type 1 and type 2 diabetes has become more difficult and very challenging, a new category of diabetes termed double or hybrid diabetes has been introduced. Double diabetes possesses symptoms of both type 1 and type 2 diabetes including obesity, insulin resistance, type of latent autoimmune diabetes in youth (LADY) autoantibodies (namely GAD56, IA2), and insulin antibodies [12].

Gestational diabetes mellitus (GDM) is defined as any degree of glucose intolerance which is first recognized during pregnancy [13]. Deterioration of glucose tolerance has its onset in the third trimester, particularly in women with marked obesity, personal history of GDM, glycosuria, or a strong family history of diabetes [14].

Defect in insulin secretion and/or resistance to insulin action in various tissues such as muscle, liver, and adipose tissue results in abnormalities in carbohydrate, fat, and protein metabolism. Hyperglycemia, as a common feature of uncontrolled diabetes, over time results in serious damage, dysfunction, and failure of various organs, such as eyes, heart, liver, kidneys, brain, nerves, and blood vessels (Figure 1), leading to long-term complications of diabetes [1].

Diabetic complications can be classified broadly as microvascular (neuropathy, nephropathy, and vision disorders) or macrovascular diseases (heart disease, stroke, and peripheral vascular disease) [15]. Diabetes and elevated blood glucose level are associated with a wide range of cardiovascular diseases that collectively comprise the largest cause of both morbidity and mortality among diabetic patients [16]. Both diabetes and cardiovascular diseases are strongly associated with chronic kidney disease, causing 80% of end-stage renal disease globally. Other complications of diabetes include infections, metabolic difficulties, erectile dysfunction, and autonomic neuropathy [17].

Diabetes and diabetes-related complications decrease life quality and present huge financial burdens for individuals and families, as well as for the health systems and economies of countries. There are many national and worldwide diabetes prevention programs focusing on preventing or delaying the onset of diabetes and its complications. While there is no effective intervention to prevent type 1 diabetes, there is strong evidence that primary prevention of type 2 diabetes can be effective. The most efficient strategy for prevention or, at least, delay of type 2 diabetes onset/development is feasible in many ethnic groups by lifestyle modification or administration of some pharmacological agents [18,19].

### Contemporary Drug Therapy for Diabetes and Major Adverse Effect

Adequate glycemic control is a fundamental part of the management of diabetes mellitus. Glycemic control is required to prevent acute symptoms and complications of hyperglycemia and to prevent or reduce and delay chronic microvascular and macrovascular complications. Current treatment protocols for management of hyperglycemia recommend an individualized approach as a part of a comprehensive management program to address coexistent disease and modifiable cardiovascular risk factors. It is emphasized that lifestyle measures such as diet and exercise should be introduced from the time of diagnosis because these measures can provide valuable blood glucose lowering efficacy [20]. However, if lifestyle intervention does not achieve adequate glycemic control, pharmacologic therapy should be introduced promptly.

In the case of type 1 diabetes the insulin replacement therapy is necessary and involves multiple daily injections of insulin or continuous subcutaneous insulin infusion. The main adverse reaction of insulin treatment is hypoglycemia, especially in intensified treatment where insufficient calorie uptake or physical exertion could be possible triggers. Additional medications also may be prescribed for people with type 1 diabetes, including medications for high blood pressure, aspirin, and cholesterol-lowering drugs. Besides taking insulin on a daily basis, patients with type 1 diabetes should be disciplined in carbohydrate, fat, and protein counting, persistent in eating healthy foods, and exercise regularly to maintain a healthy weight.

On the other hand, there are several classes of the oral hypoglycemic drugs for type 2 diabetes that point to the various targets in order to restore glucose homeostasis. The main tissues through which oral hypoglycemic drugs exert their glucose-lowering effect are pancreas, liver, skeletal muscle, adipose tissue, and intestine. According to the mechanisms of action we can distinguish multiple drugs that stimulate insulin secretion by the pancreas, increase sensitivity of target organs to insulin and decrease the rate of glucose absorption from the gastrointestinal tract.

First-line therapy drugs for patients with type 2 diabetes are the biguanides, of which metformin is the most prescribed antidiabetic agent in most countries [21]. It is now known that the main glucose-lowering effects of metformin are reduction of hepatic glucose production, enhanced peripheral glucose uptake and utilization and decrease of intestinal glucose adsorption [22,23]. These actions contribute to improving insulin sensitivity and glucose homeostasis in a patient with type 2 diabetes. Because of its main glucose-lowering effect, which appears to be a reduction of hepatic glucose production, metformin is not sufficient to cause hypoglycemia when used as mono-therapy. The main adverse effect of metformin is abdominal discomfort and other gastrointestinal problems, including diarrhea, which can be ameliorated by taking the drug with meals. The most serious but rare adverse effect of metformin therapy is lactic acidosis [24].

One of the oldest classes of oral hypoglycemic drugs, introduced in the 1950s, is represented by the insulin secretagogues, sulfonylureas [25]. Sulfonylureas act directly on the β-cells of the islets of Langerhans to stimulate insulin secretion. They enter the β-cell and bind to the cytosolic surface of the sulfonylurea receptor 1 (SUR1), causing closure of ATP-sensitive potassium channels (KATP), depolarizing the plasma membrane, opening calcium channels, and activating calcium-dependent signaling proteins that control the contractility of micotubules and mictrofilaments that mediate the exocytotic release of insulin granules [26]. The most common adverse effects of sulfonylureas are hypoglycemia, mainly because of insulin-induced suppression of hepatic glucose production. Other common limitations of these drugs are weight gain and severe cardiovascular risk due to binding of sulfonylureas to KATP channels in the cardiomyocytes, leading to ischemic preconditioning and further to myocardial infarction [27]. Due to the association of sulfonylureas with serious life-threatening events, the current 2019 ADA guidelines conserve using these drugs as last-line therapy if all other classes of antidiabetic medications fail.

A third class of oral hypoglycemic agents is thiazolidinediones (TZDs), also known as glitazones. They modulate the expression of several genes involved in differentiation of adipocytes and enzymes involved in lipid homeostasis [28] by targeting the key adipogenic transcription factor, the nuclear receptor/transcription factor peroxisome proliferator-activated receptors γ (PPARγ) in white adipose tissue. In experimental and clinical settings, TZDs decrease insulin resistance, promote lipogenesis in peripheral adipocytes, decrease hepatic and peripheral triglycerides, decrease activity of visceral adipocytes, and decrease the ratio of leptin to adiponectin, which are two important adipokines involved in appetite control and insulin sensitivity, respectively [29]. Two most significant side-effect of treatment with TZDs are congestive heart failure and risk of bone fracture. Less dramatic is the weight gain/water retention and edema that TZDs may induce. New data about TZD-mediated adverse effects improve the clinician’s ability to select patients that will have minimal significant side-effects [29].

Alpha-glucosidase inhibitors are a class of oral glucose lowering drugs which act by inhibiting enzymes of the intestinal epithelial lining involved in the digestion of complex sugars into smaller, easily absorbed monosaccharides [30]. They can be used alone or in combination therapy, notably to reduce postprandial blood glucose levels contributed by carbohydrates. However, by preventing complex carbohydrate digestion thus leaving some undigested carbohydrates, which are digested by colonic bacteria, alpha-glucosidase inhibitors exhibit dose-dependent gastrointestinal side-effects such as flatulence and diarrhea that can limit their use [30].

The newest class of hypoglycemic drugs is the incretin mimetics class, a group of injectable drugs for treatment of type 2 diabetes. They work by mimicking the functions of natural gastrointestinal peptide hormones (incretin) that act principally on pancreatic β-cells. Incretin mimetics delay gastric emptying, increase glucose-induced insulin secretion, inhibit the release of glucagon, and stimulate beta cell proliferation [31]. The later effect holds promise to counter the gradually failing pancreatic functional mass characteristic of type 2 diabetes. The most common side-effects reported are headache, nasopharyngitis, and upper respiratory tract infection [32]. Major safety concerns with incretin-based therapies include the effect of these drugs on pancreatic and thyroid tissue, since animal studies have indicated an association of these drugs with pancreatitis, pancreatic cancer and thyroid carcinoma [33].

Administration of insulin and oral and injectable hypoglycemic agents is the mainstay of treatment of diabetes and is effective in controlling hyperglycemia. Nevertheless, they have harmful side-effects, including hypoglycemia, development of insulin resistance, severe cardiovascular risks, and cancer-associated risks, and also fail to significantly alter the course of diabetic complications [2]. Thus, there is an urgent requirement for effective substitutions to reduce the complications of diabetes with fewer side-effects. In recent years, we have witnessed a renewal of attention for alternative medications and natural therapies derived from medicinal plants and mushrooms that have been used as traditional medications for thousands of years [34]. The greatest Egyptian medical document, the Papyrus Ebers of 1550 BC, represents the earliest recorded document on treatments for diabetes; it recommended a high-fiber diet of wheat grains and ochre. Due to the numerous reports and findings on the health benefits of mushrooms to humans, numerous mushroom extracts and isolated substances have been examined for antidiabetic activity, with a view to identifying alternative treatment strategies for diabetes.

## 3. Major Bioactive Components of Mushrooms in the Treatment of Diabetes

Mushrooms, filamentous fungi with fruiting bodies, are a rich source of nutrients, especially protein and carbohydrates. They are an excellent source of minerals, e.g., phosphorus, magnesium, selenium, copper, and potassium, vitamins, e.g., B vitamins and vitamin D, and essential amino acids, which are necessary for the proper functioning of the body [35,36]. Mushrooms have been considered as an essential part of the human diet because they can be used as both food and medicine due to their potential to reduce the risk of some diseases and ability to act as antibacterial, antiviral, antioxidant, antidiabetic, anticancerous, and hypocholesterolemic agents [37]. While numerous species of mushrooms exist in nature, only a few are used and cultivated as edibles. Mushrooms are considered one of the delicious foods that are easy to cultivate because they require low resources and area, and can be grown all over the world. The most cultivated edible mushrooms worldwide are *Agaricus bisporus* (common mushroom), *Lentinus edodes* (shiitake mushroom), *Pleurotus* spp. (in particular oyster mushroom), and *Flammulina velutipes* (enoki mushroom) [35,38].

Given that diabetes is characterized by elevated blood glucose levels, following a healthy diet that helps control blood glucose is essential for treatment of diabetic patients. Despite varying appearance and taste, mushrooms have similar nutritional profiles characterized by low sugar and fat content and high amounts of selenium and certain B vitamins. Due to the fact that they represent a low calorie food with a low glycemic index, they can be considered as an excellent nutrition choice for diabetic patients.

Mushrooms possess medicinal properties due to the presence of different types of secondary metabolites such as polysaccharides, lectins, lactones, terpenoids, alkaloids, antibiotics, and metal-chelating agents [39]. These secondary metabolites are bioactive compounds with a great potential to be applied as therapeutic agents. Traditionally, the bioactive components were mostly obtained from field-cultivated mushrooms. This production system was and still is a time-consuming and labor-intensive process with a very low control of the product quality and the productivity of desired metabolites [40]. While all parts of a mushroom can be used for medicinal purposes, the bioactivity is much higher in mycelia than in fruiting bodies and spores. Therefore, submerged cultivation of mushroom mycelia is a promising technology for the efficient large-scale production of mycelia biomass and value-added secondary metabolites in a compact space, shorter times, and reduced contamination [41]. Bioactive metabolites are influenced by different culture conditions such as physical conditions (temperature, pH, oxygen level, incubation time, etc.), medium composition (carbon source, nitrogen source, different salts, special additives like vegetative oils, vitamin), mode and methods of fermentation (agitated culture, static culture) [42]. The principle aim for optimization of culturing conditions is to accelerate mycelia growth and enhance productions of secondary metabolites, especially polysaccharides and triterpenoids as the most active ingredients of mushrooms [43,44,45]. This technology is feasible for actual application, because by acting on the main factors affecting the fermentation process and the purification systems, highly efficient production of various secondary metabolites can be developed. Analyzing the published results related to the use of isolated compounds and extracts derived from various mushroom species with an anti-hyperglycemic effect, it can be concluded that two groups of compounds are most important: Polysaccharides and terpenoids.

## 4. Blood Glucose-Lowering Mechanisms of Polysaccharides

Polysaccharides are complex carbohydrates, composed of the monosaccharide unit and linked by glycosidic bonds. Based on the monosaccharide composition, they are divided into homopolysaccharides and heteropolysaccharides. In both types of polysaccharides, monosaccharides can link in a linear fashion or they can branch out into complex formation. Earlier studies showed that biologically active polysaccharides are widespread among higher basidiomycetous mushrooms, and most of them belong to the group of beta-D-glucans. β-glucan, a type of dietary fiber present in sources like cereal grains, yeast, mushrooms, and prebiotic bacteria, is helpful in the prevention and control of obesity, cardiovascular diseases, diabetes, and cancer [46,47]. Besides the fact that β-glucan improves hyperglycemia, it has been shown that β-glucan administration under diabetic conditions promotes a systemic improvement which can increase the organism’s resistance to the onset of diabetic complications [48,49,50]. The fungal sources that contain β-glucan are the edible mushrooms such as *Lentinus edodes*, *Agaricus blazei*, *Schizophyllum commune*, *Ganoderma lucidum*, *Agaricus brasiliensis*, *Pleurotus florida*, and *Lentinus squarrosulus* [51]. Mushroom β-glucans are non-starch polysaccharides with a glucose polymer-chain core with beta-(1–3) linkages in the main chain of glucan and additional beta-(1–6) branch points. The core chain lengths of β-glucan differ, as do the types and complexity of side chain branching. It has been proposed that high molecular weight glucans with a higher degree of structural complexity appear to be more effective than those of low molecular weight. Mushrooms also contain different types of heteropolysaccharides β-D-glucans with chains of xylose, mannose, galactose, and uronic acid and glycoproteins β-D-glucan-protein complexes [52]. These β-D-glucan heteroglucans and protein complexes can be extracted from *G. lucidum*.

At present, a considerable number of fungi, including higher basidiomycetes, are known for their ability to synthesize exopolysaccharides (EPSs) in laboratory culture systems [42]. Major advantages of EPSs over intracellular and cell wall polysaccharides include huge production in short time, easy isolation, and purification.

Bearing in mind that dietary polysaccharides and EPS purified from mushrooms represent main bioactive compounds, which can regulate glucose homeostasis and reduce the complications of diabetes, we summarized the literature based on the anti-hyperglycemic mechanisms of their action (Figure 2).

### 4.1. Inhibiting Glucose Absorption Efficacy

Polysaccharides could attenuate diabetes by the mechanisms of gastrointestinal viscosity, inhibiting glucose absorption efficacy and postprandial glycaemia. The water-soluble dietary fibers and polysaccharides increase the viscosity of gastrointestinal content, thereby decreasing the gastric emptying rate and delaying food digestion and absorption of carbohydrates [53]. In addition, there are also indications that polysaccharide can bind and adsorb glucose, thus maintaining a low glucose concentration in the small intestine [54]. Polysaccharides from *Agaricus campestris* [55] and the EPSs of *Coriolus versicolor*, *Cordyceps sinensis*, *Paecilomyces japonica*, *Armillariella mellea*, and *Fomes fomentarius* [56] due to their hydrosolubility may reduce nutrient movement toward the villi network for efficient absorption through the increased viscosity of intestinal content, consequently reducing the glycaemia. The plasma glucose level was significantly reduced after the oral administration of EBP obtained from *C. vesicular* (29.9%), followed by *P. japonica* (21.4%), *C. sinensis* (21.2%), *F. fomentarius* (21.2%), and *A. mellea* (12.3%), as compared to control group [56].

### 4.2. Enhancement of Pancreatic β-Cell Mass

Endocrine pancreas has a significant capacity to adjust to changes in insulin demand [57], and it contains quiescent cells that can proliferate and replace dysfunctional/dead cells [58]. Therefore, investigation of different mushrooms with stimulatory effect on pancreatic β-cell regeneration represents a vital goal in the development of effective nutrient-based treatments for diabetes [59]. One of the reported mechanisms responsible for the increase in the number of functional insulin-positive β-cells is linked with the increased expression of the chemokine CXCL12 protein that mediates the restoration of β-cell population through the activation of the serine/threonine-specific Akt protein kinase prosurvival pathway [60]. This mechanism is particularly important during the initial stage of diabetes development when it is potentially possible to expand still existing β-cell mass through regeneration.

Moreover, decreased β-cell mass along with elevated β-cell apoptosis is a relatively common feature of type 2 diabetes and is directly linked to the elevated expression of Bax and decreased level of Bcl-2. Several animal studies reported that these changes, observed in the pancreas of diabetic animals, were effectively reversed after polysaccharide supplementation, and were accompanied by an elevated ratio of Bcl-2/Bax [61,62]. Polysaccharides with a capacity to induce a cell’s proliferation were obtained from *Ganoderma atrum* [62] and after its administration in diabetic animals histopathological studies showed elevated β-cell mass, pancreatic islets expansion, and restoration [63]. Another protein-bound polysaccharide from fruit bodies of *G. lucidum* exhibited similar anti-diabetic potential by inhibiting the β-cell apoptosis in streptozotocin (STZ)-induced diabetic rats [64]. The underlying mechanism initiated by *G. lucidum* is related to significant up-regulation of Bcl-2 and down-regulation of Bax and caspase in the pancreatic cells compared to that of STZ diabetic animals [65].

### 4.3. Increase of Insulin Signaling Pathways

Under physiological conditions, insulin secreted immediately after a meal activates the IRS/phosphoinositide 3-kinase (PI3K)/Akt signaling pathway. Akt is a main mediator that activates the most biochemical mechanism in the glucose metabolism via activation of phosphofructokinase and deactivation of glycogen synthase kinase 3 (GSK-3) by an increase in glucose utilization and reduction of gluconeogenesis in liver and muscle [66]. Moreover, the IRS/PI3K/Akt signaling pathway increases body lipid deposition, increases insulin production in the pancreas, and regulates lipid and glucose metabolism. However, in type 2 diabetes the PI3K/Akt pathway is damaged in various tissues as the result of insulin resistance, and in turn insulin resistance exacerbates the PI3K/Akt pathway, forming a vicious circle. GSK-3 protein expression and kinase activity are elevated in diabetes, while selective GSK-3 inhibitors have shown promise as modulators of glucose metabolism and insulin sensitivity. Administration of polysaccharides triggers insulin signaling pathways through insulin receptors, and activates the PI3K/Akt pathway by elevating the expressions of the insulin receptor (IRS1), PI3K, and Akt in type 2 diabetic animal models [62,67].

An in vitro study on the HepG2 cell model showed that the polysaccharide from *Grifola frondosa* significantly increases glucose metabolism and stimulates the synthesis of intracellular glycogen through the Akt/GSK-3 pathway. Polysaccharide from *G. frondosa* activated IRS and increased Akt expression, which leads to an inhibition of GSK-3 [68].

Polysaccharide were derived from *G. atrum* (PSG-1) upregulated mRNA expression glucose transporter-4 (GLUT4), PI3K, and phosphorylated-Akt (p-Akt) in the liver of diabetic rats. These results suggested that administration of PSG-1 in type 2 diabetic rats regulates hepatic glucose uptake by inducing GLUT4 translocation through PI3K/Akt signaling pathways [69].

Oral administration of *Inonotus obliquus* polysaccharides in a high fat diet and STZ-induced type 2 diabetic mice significantly reduced fasting blood glucose levels and up-regulated protein expressions of PI3K-p85, p-Akt (ser473) in liver, and GLUT4 in adipose tissue. These results indicated that the anti-hyperglycemic mechanism of *I. obliquus* polysaccharides involved the activation of insulin signaling through PI3K/Akt phosphorylation and the improvement of the glucose transportation by elevating the expression of GLUT4 in adipose tissues in diabetic mice [70]. Administration of total polysaccharides extracted from *Pleurotus ostreatus* in a high fat diet and STZ-induced type 2 diabetic rats for four weeks reduced hyperglycemia improved insulin resistance and increased glycogen storage by activating GSK-3 phosphorylation in liver and GLUT4 translocation in muscular tissue [71].

## 5. Blood Glucose-Lowering Mechanism of Terpenoids

Terpenes are known as an important bioactive metabolite produced by many higher fungi. They consist of multiple isoprene units (containing five carbon atoms) and are usually grouped according to the number of isoprene (C_5_H_8_) units in the monoterpenes, sesquiterpenes, diterpenes, sesterpenes, triterpenes, tetraterpenes, and politerpenes [72]. The term terpene is often extended to the terpenoids, which are oxygenated derivatives of these hydrocarbons. Diterpenoids, triterpenoids, and sesquiterpenoid are the typical representatives of terpenes with interesting biological activities. Triterpenoids and sesquiterpenoids are mainly common among mushroom metabolites and some of them have homologies with plant terpenoid compounds. Due to their lipophilic nature, triterpenes can bind to cell membranes, thereby affecting their fluidity, which in turn may limit their bioavailability. While they are large molecules, experiments revealed that triterpenoids penetrate both cell membranes and the blood–brain barrier and accumulate in the liver, circulatory system, and other tissues [72]. Moreover, chronic intake of triterpene-rich natural products increases their bioavailability and accumulation in circulation and tissues [73]. Previous studies of the antidiabetic properties of triterpenoids have been associated with inhibition of enzymes actively involved in glucose metabolism, such as aldose reductase and α-glucosidase [74,75].

### 5.1. Inhibition of α-Glucosidase

Managing postprandial hyperglycemia is a main beneficial strategy for the management of diabetes. Dietary carbohydrates are naturally digested into monosaccharides, such as glucose and fructose, which can be readily absorbed by the small intestine and transferred into the blood circulation. One of the main carbohydrate-digestive enzymes located in the small intestine epithelium, α-glucosidase, is vital for conversion of disaccharides and oligosaccharides into glucose. Hence, restraint of α-glucosidase notably inhibits the conversion of polysaccharides into blood glucose, which serves as an effective step to control postprandial blood glucose levels [76] and represents an effective strategy in diabetes for controlling the blood glucose level.

Triterpenoids are the main bioactive constituents among the compounds present in the genus *Ganoderma*, with over 140 isolated triterpenoids from fungi of this genus. Interestingly, as many as 15 triterpenoids have been identified in *G. lucidum* [74] and these isolated triterpenoids are a class of naturally occurring compounds and structurally highly oxidized lanostanes [77]. These lanostane-type triterpenes have attracted considerable attention due to their potentially significant pharmacological activities and structural diversity. These compounds showed inhibitory effects against α-glucosidase enzyme activity (Figure 3) [78]. Moreover, methanol extract of *G. resinaceum* showed inhibitory activity against α-glucosidase [79]. Analysis of forty-eight triterpenes isolated from ethanol extract of the fruiting bodies of *G. resinaceum* showed that Resinacein C, ganoderic acid Y, lucialdehyde C, 7-oxo-ganoderic acid Z3, 7-oxo-ganoderic acid Z, and lucidadiol have strong inhibitory effects against α-glucosidase compared with the positive control drug acarbose, commonly used to lower blood glucose levels after a meal. The structure–activity relationships of *Ganoderma* triterpenes on α-glucosidase inhibition showed that the C-24/C-25 double bond is necessary for α-glucosidase inhibitory activity. Moreover, the carboxylic acid group at C-26 and the hydroxy group at C-15 play important roles in enhancing inhibitory effects of these triterpenes [80].

Triterpenoid ganoderol B identified in the chloroform extract of *G. lucidum* showed greater inhibitory effects against α-glucosidase than acarbose [81] due to the presence of hydroxyl groups at C-3 and at C-25 and a double bond in the side chain which are responsible for the high activity of this compound [82]. Analysis of 25 triterpenes isolated from ethanol extracts of *G. lingzhi* fruiting bodies at various stages of development showed stronger inhibitory effects against α-glycosidase than the standard inhibitor used [83,84].

Lanostane-type triterpenoids isolated from the submerged culture of chaga mushroom, *I. obliquus*, were tested for the presence of inhibitory effect on α-glycosidase activity. Inotolactones A and B, examples of lanostane-type triterpenoids bearing α,β-dimethyl, α,β-unsaturated δ-lactone side chains, exhibited more potent α-glucosidase inhibitory activities than the positive control, acarbose. This finding might be related to the anti-hyperglycemic properties of the fungus and to its popular role in the treatment of diabetes [85].

### 5.2. Insulin Sensitizers

Activation of PPAR-γ, ligand-activated transcription factors of the nuclear hormone receptor superfamily, causes insulin sensitization and enhances glucose metabolism. PPARs function as heterodimer in association with the co-activator complex that binds to the DNA sequence present in the promoter of target genes, which leads to their transactivation or transrepression [86]. PPAR-γ enhances the expression of a number of genes encoding proteins involved in glucose and lipid metabolism [87]. In the absence of the ligands, these heterodimers are associated with a co-repressor complex which blocks gene transcription. Thiazolidinediones (TZDs) are the most widely studied PPAR-γ ligands. Recent studies show that the ability of TZD to bind and activate PPAR-γ correlate with their ability to reduce hyperglycemia in animal models of type 2 diabetes and obesity [88,89]. Bearing in mind TZD side-effects, finding novel insulin sensitizers without TZD-like side-effects will be invaluable to diabetic patients [90]. Lanostane-type triterpenes, such as pachymic acid and dehydrotrametenolic acid obtained from *Poria cocos* were analyzed for their ability to activate PPAR-γ in vitro. Transactivation experiments performed using transfected NIH3T3 cells that express PPAR-γ and PPAR-response element containing a reporter construct showed that these terpene compounds activate the reporter construct in a dose-dependent manner [91]. Moreover, application of lanostane-type triterpene dehydrotrametenolic acid reduced hyperglycemia in mouse models of type 2 diabetes and acted as an insulin sensitizer (Figure 3) as indicated by the results of the glucose tolerance test, which makes it a promising candidate for a new type of insulin-sensitizing drug [91].

## 6. Mushrooms as Drug Therapy for Diabetes—Overview of Clinical Trials

Nowadays, mushrooms are consumed as medicines or as foods in the form of dietary supplements [92]. The enormous potential of mushroom polysaccharides and terpenoids in the treatment of diabetes has been discussed in this review. However, there are still a lot of limitations and unknowns concerning the use and consumption of mushroom bioactive compounds in diabetic patients. Currently, in contrast to the high number of experimental results, there is a very limited number of clinical trials of several mushroom species that are used as drugs for patients with type 2 diabetes. In a randomized, double-blind, placebo-controlled clinical trial, the mushroom *Agaricus Blazei Murill* in combination with metformin and gliclazide improved insulin resistance among treated subjects with type 2 diabetes when compared with the placebo group [93]. Other human clinical studies aiming to evaluate the efficacy and safety *G. lucidum* in patients with confirmed type 2 diabetes with intervention lengths of 4–12 weeks have reported improvements in glycosylated haemoglobin (HbA1c), fasting plasma glucose, postprandial glucose, insulin, and C-peptide [94], whilst other studies have reported no changes in glucose parameters [95,96]. Randomized controlled trials aimed to determine the oral hypoglycaemic effect of suspensions of freeze dried and powdered *Pleurotus ostreatus* and *Pleurotus cystidiosus* showed a significant reduction in fasting and postprandial serum glucose levels of healthy volunteers and reduced the postprandial serum glucose levels and increased the serum insulin levels of type 2 diabetic patients [97]. A majority of available clinical trial reports focused on the study of crude mushroom extract or polysaccharide-enriched fractions. So, further exploration of the relevance of pure mushroom compounds and their therapeutic effects in diabetic patients is warranted.

## 7. Conclusions

Diabetes represents a serious health issue worldwide with a significant rise in morbidity and mortality every year. While the commercial hypoglycemic agents are effective in controlling hyperglycemia, they have harmful side-effects, high cost, and may cause serious complications, such as hypoglycemia, development of insulin resistance, severe cardiovascular risks, and cancer-associated risks. Therefore, the search for active antidiabetic agents from natural sources represents an exciting opportunity for the development of new types of therapeutics. Traditional medicine, together with various experimental studies, clearly indicates a great therapeutic potential of edible and medicinal mushrooms for treatment of diabetes. Nowadays, it is becoming increasingly important to isolate and identify specific bioactive compounds with antidiabetic properties and to clarify the mechanisms of their hypoglicemic action. A large number of polysaccharides and triterpens have been extracted from a wide variety of mushroom species and their structural features and mechanisms of hypoglycemic activities have been elucidated. The bioactivity mechanisms, biosynthetic pathways, and productivities of the polysaccharides and triterpenes are also highly variable in different mushroom species and different cultivation conditions. Bearing in mind that there is a clear relationship between the structure and bioactivity of these secondary metabolites, continued investigation along this line will help to identify the most effective structures, and improve regulation strategies in a controllable biosynthesis of polysaccharides and triterpenes with enhanced bioactivities. Furthermore, well-designed randomized controlled trials with long-term consumption are needed to guarantee the bioactivity and safety of mushroom products for diabetic patients.

## Figures and Tables

**Figure 1 jof-07-00058-f001:**
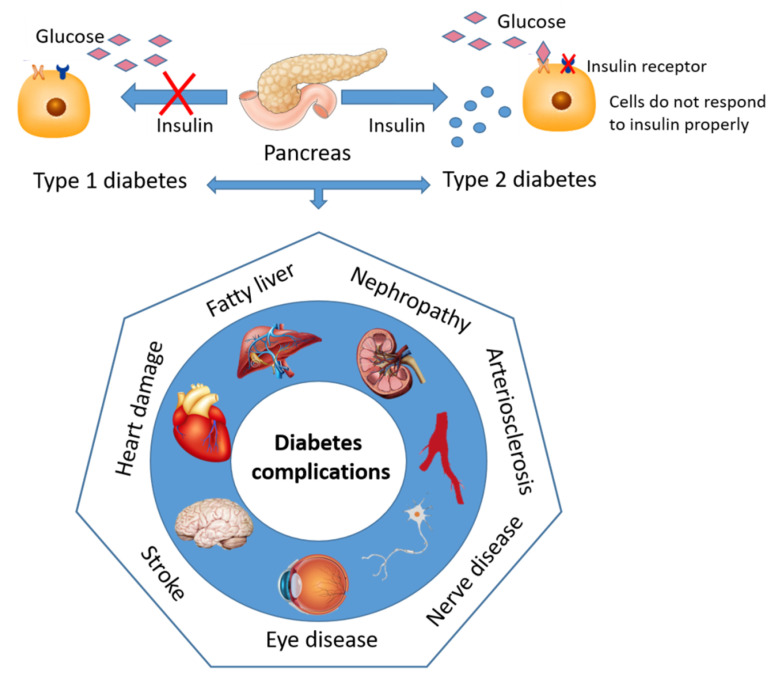
Major microvascular and macrovascular complications associated with diabetes mellitus.

**Figure 2 jof-07-00058-f002:**
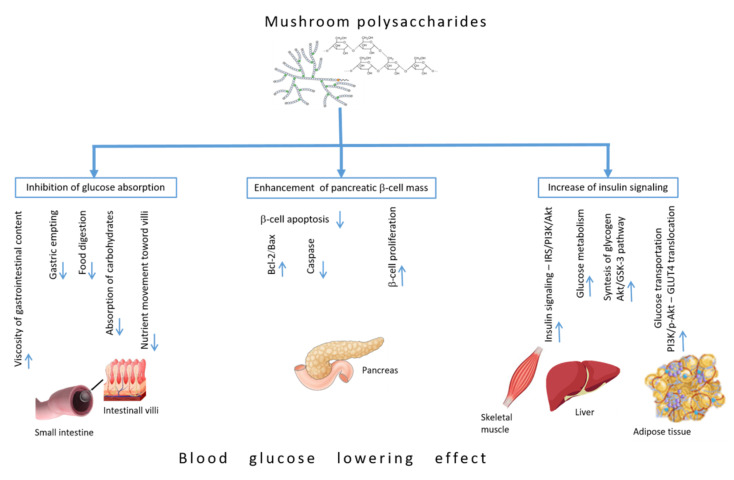
Anti-hyperglycemic mechanisms of mushroom polysaccharides in different tissues involved in glucose homeostasis. IRS: Insulin receptor substrate; PI3K: Phosphoinositide 3-kinase; AKT: Serine/threonine-specific protein kinase; GSK-3: Glycogen synthase kinase-3; GLUT4: Glucose transporter-4.

**Figure 3 jof-07-00058-f003:**
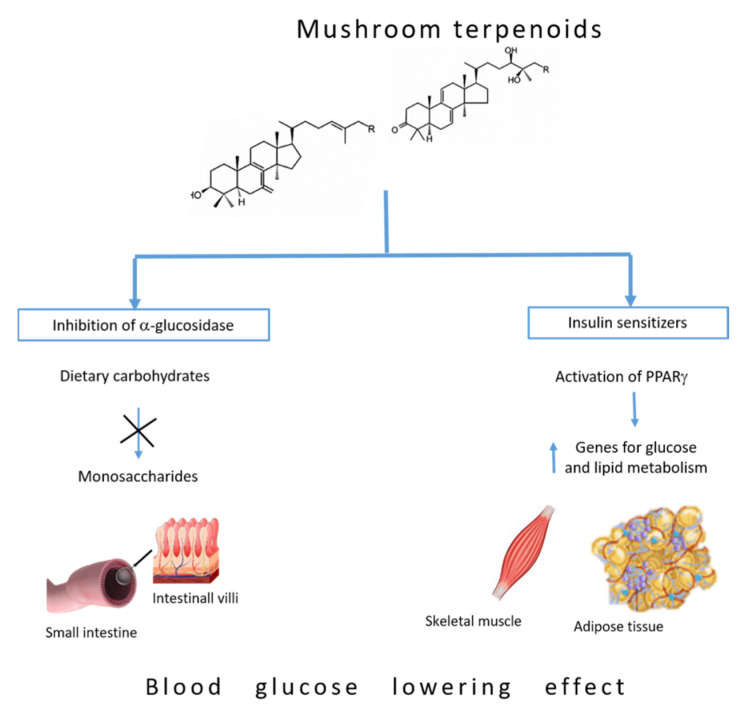
Anti-hyperglycemic mechanisms of mushroom terpenoids in different tissues involved in glucose homeostasis. PPAR-γ: Peroxisome proliferator-activated receptors γ.

## Data Availability

This review did not report any unpublished studies.

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
