# Peer review of "The Effects of Major Mushroom Bioactive Compounds on Mechanisms That Control Blood Glucose Level"

_jof, 2021, doi:10.3390/jof7010058_

Round 1

Reviewer 1 Report

The subject and overall content of the manuscript is very interesting and highlight the importance of mushroom bioactive compounds on controlling blood sugar level.  The manuscript is well-structured and clearly written.  This manuscript can be published in the present form, though it could be improved by making some critical comments on the reported work, which would significantly contribute to the existing body of knowledge.

Author Response

We thank the Reviewer 1 for the stimulating comment in favor of publishing.

Reviewer 2 Report

The current review paper describes major bioactive molecules from medicinal fungi that could be valuable source of anti-diabetic agents. This review paper is of good quality, it is well organized and the most important findings concerning recent literature were presented. Therefore, I suggest minor revisions before acceptance of the paper.

Since this is a review paper, could you please state how did you search for the most recent findings on this subject? (eg. keywords entered in the search tab at web of science/scopus/google scholar). Can you indicate which years of publications are considered for this review article regarding compounds from fungi? (eg. literature data from 2010-2020).

Have you performed meta analysis in order to select the most important papers in the filed regarding anti-diabetic molecules from fungi? If not, how did you choose to present these findings; is the current literature knowledge scarce on this subject?

Are there any human clinical trials on the use of compounds derived from fungi in diabetic patients? You have concluded that further clinical trials are necessary and I agree, but there should be at list a few reports available (eg. https://doi.org/10.1038/srep29540, and take a look at this one https://doi.org/10.1615/IntJMedMushrooms.2020035863).

You should correct typographical mistakes through the manuscrpt, eg .. etc.

Latin names of fungi should be italicized all over the manuscript.

Nice work and easy to read and catch the point, congratulations! 

Author Response

The current review paper describes major bioactive molecules from medicinal fungi that could be valuable source of anti-diabetic agents. This review paper is of good quality, it is well organized and the most important findings concerning recent literature were presented. Therefore, I suggest minor revisions before acceptance of the paper.

Since this is a review paper, could you please state how did you search for the most recent findings on this subject? (eg. keywords entered in the search tab at web of science/scopus/google scholar). Can you indicate which years of publications are considered for this review article regarding compounds from fungi? (eg. literature data from 2010-2020).

Our response: The aim of this review was to summarize the literature data regarding the underlying molecular mechanisms of main mushroom secondary metabolites (polysaccharides and terpenoids) responsible for lowering blood glucose levels. Accordingly, the key words for relevant article search were mushroom secondary metabolites, anti-diabetic properties, mushroom polysaccharides, mushroom terpenoids, anti-hyperglycemic effect, mechanism of action etc. combined in different ways at ScienceDirect and PubMed in most cases. This publication search was limited to the period from 2010 to 2020.

Have you performed meta analysis in order to select the most important papers in the filed regarding anti-diabetic molecules from fungi? If not, how did you choose to present these findings; is the current literature knowledge scarce on this subject?

Our response: The meta-analysis has not been performed. We decided to go for a so-called narrative review with attempt to gather all available empirical research since, meta-analysis could offer statistically produced analysis and combination of results from several similar studies that does not fit to our final aim. Bearing in mind that we set the aim of this review very narrow, it was very challenging to find literature that will meet the pre-set criteria. Although there are many research articles dealing with the anti-diabetic properties of various mushrooms species, most of them are related to the examination of the effects of the total extract and not the individual compounds. If we further narrow the search to those studies that deal not only with the ascertainment of the anti-diabetic effect of the isolated compound, but also with its mechanism of action, then we come to the situation that the number of such studies is rather limited. Because of that we performed this narrative review in order to provide comprehensive narrative synthesis of evidence based on currently available literature and furthermore, we overcome the common criticism of meta-analysis i.e. combining different kinds of studies in the same analysis.

Are there any human clinical trials on the use of compounds derived from fungi in diabetic patients? You have concluded that further clinical trials are necessary and I agree, but there should be at list a few reports available (eg. https://doi.org/10.1038/srep29540, and take a look at this one https://doi.org/10.1615/IntJMedMushrooms.2020035863).

Our response: As suggested by Reviewer 2, we provided an additional report regarding human clinical trials on the use of mushrooms in the treatment of diabetes which has been included in the manuscript in the paragraph 6. Mushrooms as drug therapy for diabetes-overview of clinical trials.

You should correct typographical mistakes through the manuscrpt, eg .. etc.

Our response: Each noted typographical mistake has been corrected through the manuscript.

Latin names of fungi should be italicized all over the manuscript.

Our response: Latin names of fungi are italicized (due to the formatting all italicized terms somehow disappeared in the previous version of manuscript).

We thank the Reviewer 2 for the stimulating comment in favor of publishing.

Reviewer 3 Report

All Latin names of mushrooms should be written in italics, see attached file where are marked in yellow

Author Response

Latin names of fungi are italicized (due to the formatting all italized terms somehow disappeared in the previous version of manuscript).

We thank the Reviewer 3 for the stimulating comment in favor of publishing.